

# Ecosystem feedbacks from subarctic wetlands: vegetative and atmospheric CO₂ controls on greenhouse gas emissions

Matthew J. Bridgman, Barry H. Lomax and Sofie Sjögersten*

School of Biosciences, University of Nottingham, Sutton Bonington Campus, Loughborough, LE12 5RD UK

Correspondence to: sofie.sjogersten@nottingham.ac.uk

## Abstract

Wetland vegetation provide strong controls on greenhouse gas fluxes but impacts of elevated atmospheric carbon dioxide ($CO_2$) levels on greenhouse gas emissions from wetlands are poorly understood. This study aims to investigate if elevated atmospheric $CO_2$ enhance methane ($CH_4$) emissions from subarctic wetlands and to determine if responses are comparable or species specific within the Cyperaceae, an important group of artic wetland plants. To achieve this we carried out a combined field and laboratory investigation to measure of $CO_2$ and $CH_4$ fluxes. The wetland was a $CH_4$ source with comparable fluxes from areas with and without vegetation and across the different sedge communities. In contrast, the net ecosystem exchange of $CO_2$ differed with sedge species. Within the laboratory experiment plants grown at double ambient (800 ppm) $CO_2$, total biomass of *Eriophorum vaginatum* and *Carex brunnescens* increased, whereas the total biomass of *E. angustifolium* and *C. acuta* decreased, compared to the control (400 ppm $CO_2$). These changes in biomass were associated with corresponding changes in $CH_4$ flux. *E. vaginatum* and *C. brunnescens* mesocosms produced more $CH_4$ when grown in 800 ppm atmospheric $CO_2$ when compared to 400 ppm $CO_2$ with *E. angustifolium* and *C. acuta* producing less. Additionally, redox potential and carbon substrate availability in the pore water differed among the plant treatments and in response to the elevated $CO_2$ treatment. Together, this suggests species specific controls of $CH_4$ emissions in response to elevated $CO_2$, which facilitate differential plant growth responses and modification of the rhizosphere environments. Our study highlights species composition as an important control of greenhouse gas feedbacks in a $CO_2$ rich future, which need to be considered in models aiming to predict how ecosystems respond to climate change.

## 1. Introduction

High latitude wetlands are important global carbon (C) stores with approximately half of global soil C in found in the northern circum-polar permafrost region (Tarnocai *et al.*, 2009). These wetlands are under threat from climate warming (IPCC, 2013). Additionally, atmospheric $CO_2$ concentrations have increased from pre-industrial levels of 280 ppm to close to 400 ppm in 2013 with future atmospheric $CO_2$ concentrations predicted to increase to between 426 ppm (RCP 2.6) and 936 ppm (RCP 8.5) over the next century (IPCC, 2013). These changes in climate and atmospheric $CO_2$ concentration have the potential to increase net primary productivity (NPP) and decomposition rate and hence greenhouse gas emissions (Curtis *et al.*, 1989; Valentine *et al.*, 1994). Wetlands contribute around 80 % of the powerful greenhouse gas methane ($CH_4$) production from natural sources and make up a third of overall global emissions (Kirschke *et al.*, 2013). The largest $CH_4$ atmospheric mixing ratios are found north of 40° N



(Steele *et al.*, 1987), with the distribution of wetlands in the northern hemisphere recognised as a significant contributor to the global $CH_4$ budget (Moore and Knowles, 1990).

Emissions of $CH_4$ from natural wetlands are closely related to the temperature and hydrology of the area (Updegraff *et al.*, 2001; Bridgham *et al.*, 2013). In subarctic and arctic regions, these factors are strongly controlled by permafrost; hence future changes permafrost could impact greatly on regional $CH_4$ emissions (IPCC, 2013; Christensen *et al.*, 2004). For example, waterlogging of previously aerobic soils may increase $CH_4$ emissions from arctic regions (ACIA, 2005).

Vegetation is a primary control of $CH_4$ emissions from wetlands (Heilman and Carlton, 2001; Ström *et al.*, 2005; Bhullar *et al.*, 2013). This is in part because most of the organic matter stored in arctic peatlands is recalcitrant and unavailable for digestion by anaerobic bacteria (Bridgham *et al.*, 2013). Therefore, input of recent photosynthates in the form of litter or root exudates are an important carbon source for methanogens (Torn and Chapin, 1993; Ström *et al.*, 2005). The diffusion of oxygen through plant aerenchyma from the atmosphere into the roots and subsequent leakage into the rhizosphere leads to oxidation of $CH_4$ to $CO_2$ in the soil, substantially reducing net $CH_4$ emissions (Fritz *et al.*, 2011). The quality and quantity of plant litter and root exudate as well and root $O_2$ inputs differs among wetland plant species, potentially creating species specific impacts on $CH_4$ fluxes (Updegraff *et al.*, 1995; Ström *et al.*, 2005).

Elevated atmospheric $CO_2$ can influence $CH_4$ production through its impacts on plant C assimilation and allocation. For example, increased below ground biomass production in response to elevated $CO_2$ levels has been found to substantially increase $CH_4$ emissions from paddy rice fields (van Groenigen *et al.*, 2012). Increases in plant biomass and productivity in a range of wetland species in response to elevated $CO_2$ have been found to increase $CH_4$ emissions (Megonigal and Schlesinger, 1997; Kao-Kniffin *et al.*, 2011; Wang *et al.*, 2013) while other species have been unresponsive (Angel *et al.*, 2012). The contrasting responses found among wetlands may, in part, be controlled by the plant species composition as raised atmospheric $CO_2$ influences all aspects of plant activity including growth, photosynthetic rates and root exudate production, processes which vary strongly among species (Lawlor and Mitchell, 1991; Zak *et al.*, 1993; Bellisario *et al.*, 1999). These findings suggest that more detailed understanding is required to tease apart the controls that govern plant mediated impacts on $CH_4$ emissions in response to elevated $CO_2$. With regards to impacts of elevated atmospheric $CO_2$ concentrations on $CH_4$ fluxes, increased biomass may increase labile C inputs and hence production of $CH_4$ but also transport of $O_2$ to the rhizosphere and $CH_4$ to the atmosphere (Joabsson *et al.*, 1999; Wolf *et al.*, 2007; Laanbroek, 2010). However, our current understanding of impacts of elevated $CO_2$ on arctic wetland $CH_4$ emissions is limited at both the ecosystem and species level, creating large uncertainties in model predictions of the role of elevated $CO_2$ on $CH_4$ feedback mechanisms (Ringeval *et al.*, 2011).

This is an important knowledge gap as arctic wetlands are currently responding strongly to climate warming resulting both in expansions of graminoid-dominated flooded areas (Prater *et al.*, 2007; Åkerman and Johansson, 2008) and dramatic increases in $CH_4$ emissions (Christensen *et al.*, 2004; Hodgkins *et al.*, 2014). To explore how elevated atmospheric $CO_2$ impacts subarctic plant species and wetland $CH_4$ emissions we carried out *in situ* measurements of $CO_2$ and $CH_4$ emissions in a subarctic wetland in northern Sweden, comparing adjacent open water areas to areas vegetated by different *Carex* and *Eriophorum* species to determine variation in field $CH_4$ emissions. We then established a controlled environment experiment exposing peat mesocosms planted with *Carex acuta*, *Carex brunnescens*,





*Eriophorum vaginatum* and *Eriophorum angustifolium* respectively, to elevated $CO_2$ and quantified how this affected $CO_2$ and $CH_4$ fluxes, plant growth and peat physicochemical properties. This

combined approach was used to test the hypothesis that: Elevated atmospheric $CO_2$ will increase productivity of *Carex* and *Eriophorum* species and subsequently stimulate $CH_4$ emissions due to increased root inputs of dissolved organic carbon providing labile substrates for methanogens.

## 2. Methods


### 2.1. Site description

The study site is a subarctic wetland located on the southern edge of Lake Torneträsk in Northern Sweden (68° 21' 30.96" N 18° 46' 56.064" E). The mean annual precipitation is 310 mm, over 40% of

which occurs during summer, mean annual temperature is 0.7 $^{o}$C with a summer average of 11 $^{o}$C (1913-2000 average, Kohler *et al*., 2006). The site is a palsa mire complex made up of two distinct communities of vegetation. The raised, mesic area is dominated by dwarf shrubs (*Betula nana*, *Empetrum nigrum* and *Vaccinium uliginosum*) and these hummocks have a summer active layer depth of $30 \pm 0.9$ cm. In the flooded areas there are three dominant Cyperaceae species: *Carex acuta*,

*Eriophorum angustifolium* and *Eriophorum vaginatum* as well as the less common *Carex brunnescens*. On average, (n =5) *C. acuta* grew in locations with an active layer depth of $119 \pm 21$ cm, *E. angustifolium* at $122 \pm 12$ cm and *E. vaginatum* at $95 \pm 21$ cm. The water table depth in the flooded areas varied, averaging $34 \pm 7$ cm where *C. acuta* was found, $30 \pm 3$ cm for *E. angustifolium* and $15 \pm 2$ cm for *E. vaginatum*.


### 2.2. Experimental design and analysis

#### 2.2.1. Field campaign

Fieldwork was undertaken during July 2014. Summer precipitation in 2014 was 130 mm, slightly above average and mean temperature over the sampling period was 13.3°C. To establish the direction and extent of wetland-atmosphere carbon fluxes, *in situ* $CH_4$ and $CO_2$ fluxes were measured. Across the site, five plots were established for each of the three dominant Cyperaceae species. At each plot, gas fluxes were recorded using two 21.2 litre (15 cm diameter x 120 cm height) transparent fan-circulated

headspace chambers over both individual plants and open water as experimental pairs, located within a distance of one metre of each other. Air samples were taken at intervals of 5, 10, 20 and 30 minutes and stored in evacuated 12 ml exetainers (Labco, Lampeter, UK). Fluxes were measured twice for each of the 30 plots during the sampling week. $CH_4$ and $CO_2$ concentrations were determined by gas chromatography (GC-2014, Shimadzu UK LTD, Milton Keynes, UK) using a single injection system

with a 1 mL sample loop that passed the gas sample using $H_2$ as carrier. Thermal conductivity (TCD) and $H_2$ flame ionization (FID) detectors were used to measure $CO_2$ and $CH_4$, respectively. $CH_4$ and $CO_2$ evolution was examined for linearity. Gas fluxes were calculated using the ideal gas law (e.g. Mangalassery *et al*., 2014) and were expressed as both per unit area and peat dry weight.

#### 2.2.2. Growth room experiments

Growth room experiments were established using two chambers which had fixed atmospheric $CO_2$ concentrations of 400 ppm and 800 ppm. Vegetated and non-vegetated peat treatments were split between the chambers. Mesocosms of control peat and peat planted with either *C. acuta*, *C.*





*brunnescens, E. angustifolium* or *E. vaginatum* were established with peat and plant material collected from the field site. The degree of replication was; n =10 for *C. acuta* and *E. vaginatum*, n = 6 for *E. augustifolium*, and n =5 for *C. brunnescens,* and unplanted peat 'control' treatments (n=5). Peat samples were collected from submerged areas with a water table depth of ca. 30 cm. The recovered plant and soil samples were transported, separated and transplanted into separate water-tight one litre pots. The

conditions used in the growth chambers simulated the subarctic growing season. Day length was 16 hours, day/night temperature was $21/15\ ^oC$, daytime light levels were 400 µmol m$^{-2}$ s$^{-1}$ and day/night humidity was 65/75%.

Two types of head space chamber were used for the gas sampling. A taller chamber (15 cm diameter x

100 cm height, 17.7 litre volume) was used for the mesocosms with *C. acuta* and *E. angustifolium* and a smaller chamber (15 cm diameter x 25 cm height, 4.4 litre volume) was used for the shorter *E. vaginatum* and *C. brunnescens*.

To define individual plant-mediated methane-controlling mechanisms over the experimental period, gas

flux, redox, and plant extension growth measurements were measured fortnightly. These measurements were taken at five time points over a 10 week period between January and April 2015. Gas fluxes were determined using static headspace chambers, taking air samples ca. 2 minutes after the chamber was closed and then again after 20 minutes. Air in the chambers was circulated using small computer fans. The air samples were stored in 12 ml exetainers and analysed for $CH_4$ and $CO_2$ using gas

chromatography (as above). Redox potential of the soil was measured in three locations in each pot using a redox probe connected to a millivolt pH meter. For plant samples, three leaves of each individual were labelled and extension growth recorded. At the conclusion of the experiment, pore-water samples were extracted from each pot using rhizon samplers. From these samples, E4:E6 ratio, which is an indices of the humification capacity of dissolved organic carbon in the solution, was

determined using a spectrophotometer (Cecil CE1011 1000 series) at wavelengths of 465 nm and 665 nm (Worrall *et al*., 2002). TOC-TN analysis (Shimadzu TOC-V CPH; TNM-1) was used to measure the total dissolved organic carbon (TOC) and total dissolved nitrogen (TN) content of the water. and the ratio of TOC:TN reflects the lability of carbon in the pore water (Kokfelt *et al*., 2009). Above and below ground biomass of plant samples as well as soil organic matter was separated, dried at 60 $^oC$ for

72 hours then weighed to calculate total above and below ground biomass.

### 2.2.3. Data analysis

All data analysis was carried out using GenStat (15$^{th}$ Edition). Plant-related controls of *in situ* $CH_4$ and

$CO_2$ fluxes were analysed using linear mixed models, using species and plant/open water factors as the fixed model and block as the random model. For the *ex situ* experiment, the fixed model included the $CO_2$ treatment, species treatment and time factors. Individual pots were used as the random factor. Statistics reported are the *F*-value, which is the ratio for between group variance and within group variance, numerator (i.e. fixed) degrees of freedom, denominator (i.e. residual) degrees of freedom, the

*P*-value indicating significance when < 0.05. When required, data were transformed to meet the normality assumption. Relationships between variables (e.g. $CH_4$ and $CO_2$ fluxes, biomass, pore water chemistry) were analysed using linear regression.





## 3. Results

Species differences in $CH_4$ fluxes were not significant in the field. All sites were net sources of $CH_4$,
with the highest fluxes occurring in areas with *C. acuta* where mean emissions were $26 \pm 7$ mg m$^{-2}$ h$^{-1}$
compared to $10 \pm 3$ mg m$^{-2}$ h$^{-1}$ for those containing *E. vaginatum,* where fluxes were lowest (Fig. 1 a).
$CH_4$ fluxes did not differ significantly among open water and vegetated plots. In contrast, $CO_2$ fluxes
varied significantly among species ($F_{2, 2} = 2.89$, $P = 0.067$). *E. vaginatum* (-253 $\pm$ 150 mg m$^{-2}$ h$^{-1}$) and
*C. acuta* (-255 $\pm$ 122 mg m$^{-2}$ h$^{-1}$) vegetated plots were net sinks of $CO_2$ exhibiting negative fluxes,
whilst *E. angustifolium* plots tended to be a net source of $CO_2$ (357 $\pm$ 271 mg m$^{-2}$ h$^{-1}$) (Fig. 1 b).

In the *ex situ* experiment, both above ground biomass ($F_{1,3} = 3.58$, $P = 0.064$) and the shoot:root ratio
($F_{1,3} = 18.66$, $P < 0.001$) were significantly or near significantly affected by the $CO_2$ treatment for the
different species (Table 1). Specifically, elevated atmospheric $CO_2$ levels increased above ground
biomass in *E. angustifolium*, *E. vaginatum* and *C. brunnescens* but decreased above ground biomass in
*C. acuta*. The 800 ppm $CO_2$ treatment altered allocation of carbon in *E. angustifolium* increasing
shoot:root ratio to $1.1 \pm 0.14$ compared to $0.39 \pm 0.06$ in the ambient $CO_2$ (400 ppm) treatment.
Elevated atmospheric $CO_2$ increased total biomass production in *E. vaginatum* and *C. brunnescens* but
did not effect total biomass production in *E. angustifolium* or *C. acuta*.

The elevated atmospheric $CO_2$ treatment resulted in contrasting effects on $CH_4$ fluxes among the four
plant species treatments over time ($CO_2$ treatment × species × time; $F_{12, 12} = 2.65$, $P = 0.003$; Fig. 2 a-d).
$CH_4$ emissions were higher in the 400 ppm treatment for *E. angustifolium* and *C. acuta* although the
effect was variable over time, with fluxes showing greatest $CH_4$ releases at the beginning of the study
period for *C. acuta* while releases were greatest at the end of the period for *E. angustifolium*. In
contrast, the *C. brunnescens* treatment acted as a $CH_4$ sink at 400 ppm $CO_2$, while fluxes of $CH_4$ were
close to zero at 800 ppm $CO_2$. Elevating $CO_2$ concentrations did not alter $CH_4$ fluxes in the unplanted
control treatments. The changes in observed fluxes were similar irrespective of whether the data was
expressed as a function of unit area or dry weight of peat in the mesocosms (Fig. 5, supplementary
information).

*Ex situ* $CO_2$ fluxes (Fig. 2 b) were significantly effected by atmospheric $CO_2$ ($F_{1, 12} = 5.24$, $P = 0.026$)
and varied over time ($F_{4, 12} = 2.5$, $P = 0.044$). In *E. angustifolium*, *E. vaginatum* and *C. acuta*
treatments, $CO_2$ fluxes tended to be more positive under elevated atmospheric $CO_2$ when compared to
ambient $CO_2$ conditions, showing increased $CO_2$ source potential. Fluxes from *C. brunnescens*
demonstrated an opposite response to elevated atmospheric $CO_2$, switching from a $CO_2$ source to a $CO_2$
sink. Fluxes in the unplanted control treatment were smaller relative to the planted treatments and did
not vary significantly between $CO_2$ treatments. $CO_2$ fluxes from the *C. acuta* mesocosms were highest
at the beginning of the study period whereas releases from *E. angustifolium* mesocosms peaked at the
end of the experimental period.

Redox was consistantly higher in the unplanted control treatment compared to planted treatments ($F_{4, 16}$
$= 40.09$, $P < 0.001$, Fig. 3). In the elevated $CO_2$ treatment, *Carex* species further lowered redox
potentials (interaction between $CO_2$ × plant × time ($F_{16, 16} = 3.41$, $P < 0.001$)).

The elevated $CO_2$ treatment had a contrasting effect on total dissolved organic carbon (TOC) for the
different species (species × treatment interaction ($F_{3, 3} = 2.82$, $P = 0.048$), Fig. 4 a). TOC was highest in
the unplanted controls, followed by *E. vaginatum* and *C. acuta*. The response of these two species to



elevated $CO_2$ differed, with pore water in the 800 ppm treatment exhibiting 0.9 mg $L^{-1}$ more organic carbon in *E. vaginatum* but 1.5 mg $L^{-1}$ less in *C. acuta* when compared to ambient $CO_2$ conditions. In contrast, total dissolved nitrogen (TN) (Fig. 4 b) was not significantly influenced by treatment or species effects.

Pore water in the two species of *Carex* display the highest E4:E6 ratio (i.e. relatively more low molecular weight compounds) ($F_{3, 3} = 6.05$, $P = 0.001$, Fig. 4 d) while the TOC:TN ratio was highest in *E. vaginatum* ($F_{3, 3} = 7.91$, $P < 0.001$, Fig. 4 c) out of the planted treatments. There was no significant difference between the $CO_2$ treatments for these ratios.

### 4.    Discussion


The *in situ* $CH_4$ fluxes (Fig. 1 a) were of similar magnitude to those measured in minerotrophic mires in the Torneträsk area (Öquist and Svensson, 2002; Christensen *et al*., 2004; Koelbener *et al*., 2010). The net ecosystem exchange of $CO_2$ (NEE) was most negative (representing net $CO_2$ uptake from the atmosphere) in the dense stands of the tussock forming *E. vaginatum* and the taller and more bulky *C.*
*acuta* (Fig. 1 b). However, $CH_4$ fluxes did not vary substantially among areas dominated by different plant species, irrespective of the area being vegetated or open water (Fig. 1 a). Since our paired measurements were done relatively close to each other spatially, the lack of difference between open and vegetated areas may be due to similar level of rhizosphere stimulation of $CH_4$ production (Dorodnikov *et al.,* 2011) over small spatial scales. Additionally, as all sites were flooded, surface $CH_4$
oxidation is unlikely to cause differential net $CH_4$ emissions between vegetated areas, where $CH_4$ can be transported to plant tissues, and unvegetated areas, potentially explaining why we saw no difference in net emissions from plots with and without out plants (Laanbroek, 2010).

The contrasting responses of above ground and total biomass of the four sedge species to elevated $CO_2$
(Table 1) indicate that wetland plant species differ in their capacity to respond to atmospheric $CO_2$ levels. Species specific biomass responses after two years exposure to elevated $CO_2$ (ambient + 340 ppm) were also found in temperature salt marsh plant species (*Schoenoplectus americanus* and *Spartina patens*) (Langley *et al.,* 2013) as well as for three different *Typha* species (*T. angustifolia, T. glauca* and *T. latifolia*) exposed to 350–390 (control) to 550–600 ppm (treatment) $CO_2$ (Sullivan *et al*., 2010).
*Typha* species analysed by Sullivan *et al*., (2010) showed a uniform response with all species responding to the increase in atmospheric $CO_2$ by increasing below ground biomass. This is in contrast to our study where above ground biomass of *E. angustifolium, E. vaginatum* and *C. brunnescens* increased in response to elevated $CO_2$ and below ground biomass only increased for two of the study species (*E.vaginatum* and *C. brunnescens* (Table 1)). The limited below ground biomass responses for
two of our study species compares with the findings by Langley *et al.* (2013) who reported no significant changes in below ground biomass of *Schoenoplectus americanus* and *Spartina patens* after two years exposure to elevated $CO_2$. Species specific responses to atmospheric $CO_2$ are well known, with fundamental differences in stomatal numbers and size being observed (Woodward *et al*., 2002; Lomax *et al*., 2014), which can then influence physiology and ultimately impact on biomass.


The switch in NEE found in three (*E. angustifolium*, *E. vaginatum* and *C. acuta*) of the plant treatments (Fig. 2 b) may be caused by reduced photosynthesis rates in the elevated $CO_2$ treatment, which have been found previously for beech tree saplings grown under elevated $CO_2$ (Urban *et al*., 2014). However, this does not match with the greater above ground biomass found for *E. angustifolium* and *E. vaginatum*
under elevated $CO_2$. Furthermore, increased root and/or soil respiration rates, possibly due to the greater





root biomass as found in response to elevated $CO_2$ for *E. vaginatum* or greater levels of root exudation, (e.g. increased porewater TOC levels in the 800 ppm treatment for *E. vaginuatum*) may be the cause of the increased $CO_2$ emissions. The reduction in the $CO_2$ sink strength in response to elevated $CO_2$ in the *E. angustifolium*, *E. vaginatum* and *C. acuta* mesocosms contrast with studies suggesting that elevated

atmospheric $CO_2$ will increase the $CO_2$ sink strength of wetland ecosystems by increasing NPP (King *et al.*, 1997; Megonigal and Schlesinger, 1997; Sullivan *et al.*, 2010).

The pattern in the response of $CH_4$ production to atmospheric $CO_2$ mirror those found for biomass production (Fig. 2 a and Table 1). This suggests that the growth response of different species to elevated

$CO_2$ concentration has an impact on how these species influence $CH_4$ emission. For example, $CH_4$ emissions from *Taxodium distichum* and *Orontium aquaticum* mesocosms increased by. 65 and 28 % in response to an experimental increase in $CO_2$ levels from 350 to 700 ppm, reflecting changes in plant growth (Vann and Megonigal, 2003). Furthermore, $CH_4$ production in mesocosms planted with *Typha angustifolia* more than doubled, when $CO_2$ levels were increased from 380 to 700 ppm, due to increased

root biomass (Kao-Kniffin *et al.*, 2011). Similar large increases in $CH_4$ emissions (136 % increase) were reported for *Orontium aquaticum* when exposed to double ambient $CO_2$ concentration. However, in this study only photosynthesis, and not plant biomass, increased significantly in response to the $CO_2$ treatment (Megonigal and Schlesinger, 1997). In contrast, limited or no impact of elevated $CO_2$ on $CH_4$ emissions was found in two sedge dominated salt marsh communities (Marsh *et al.*, 2005). Limited

responses to elevated $CO_2$ by some species may be linked to nutrient limitation, indicating that global change responses of wetland $CH_4$ emissions may be strongly controlled by the nutrient demands of species and site nutrient status (Mozdzer and Megonigal, 2013).

In our study, the different plant species treatments controlled the amount and quality of substrate found

in the pore water, with elevated atmospheric $CO_2$ influencing TOC concentrations in planted treatments (Fig. 4a), largely reflecting trends in biomass (Table 1). This is in line with findings from temperate salt marshes exposed to elevated $CO_2$ (Marsh *et al.,* 2005; Keller *et al.,* 2009). The contrasting porewater chemistry with regards to the E4:E6 and TOC:TN ratios (Fig. 4 c and d) highlights the influence of species composition in these wetlands on rhizospheric carbon inputs, likely due to differences in the

composition and quality of root exudates, with implications for $CH_4$ production (King *et al.*, 2002; Ström *et al.*, 2005; Dorodnikov *et al.*, 2011). In addition, it has also been shown that root exudates can stimulate decomposition of more recalcitrant soil organic matter (Basiliko *et al.*, 2012). The length of our experimental period was too short to measure the effect of litter inputs. However, any observed increases in biomass as a result of raised atmospheric $CO_2$ (namely in *C. brunnescens* and *E. vaginatum,*

Table 1) is expected to increase labile carbon inputs from litter production which may further stimulate $CH_4$ production (Curtis *et al.*, 1990).

The presence of vegetation was also found to lower redox potential (Fig. 3), a critical control of $CH_4$ production (Bridgham *et al.*, 2013), which compares with findings from mesocosms with *Phragmites*

*australis* grown under ambient elevated $CO_2$ (+330 ppm $CO_2$) (Mozdzer and Megonigal, 2013). We suggest that the redox-reducing potential of plant roots is due to increased provision of labile substrate for microbial respiration, depleting alternative electron donors in the micropores where $CH_4$ production takes places (Yavitt and Seidman-Zager, 2006; Laanbroek, 2010). Our findings and those of Mozdzer and Megonigal (2013) contrast with those of Wolf *et al.* (2007) who demonstrated higher soil redox

potentials in mesocosms planted with *Scirpus olneyi* due to greater root $O_2$ inputs reflecting greater root biomass in the elevated $CO_2$ treatment. Differential impacts on the soil redox status by plants exposed to high $CO_2$ levels may strongly alter biogeochemical cycling in soils including $CH_4$ production and





oxidation rate (Fritz *et al*., 2011), potentially explaining a proportion of the variable responses observed in plant mediated changes in $CH_4$ emissions due to elevated atmospheric $CO_2$.


In conclusion, we have demonstrated that elevated atmospheric $CO_2$ increased total biomass production in *E. vaginatum* and *C. brunnescens* but not in *E. angustifolium* and *C. acuta*. In parallel to this, elevated $CO_2$ only increased $CH_4$ emissions from the *E. vaginatum* and *C. brunnescens* treatments. These data suggest a link between increased productivity via $CO_2$ fertilisation that could drive changes

in species composition, which may ultimately lead to an increase in wetland $CH_4$ emissions. Our results highlight the need for improved mechanistic understanding, at the species level, of wetland plants to elevated $CO_2$ before assumptions can be made with regards to impacts on elevated $CO_2$ on greenhouse gas emissions from wetlands.

**Acknowledgements:** We are grateful to James Verran and John Corrie for laboratory for technical support. The project was funded by the University of Nottingham.

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



**Figures and tables**

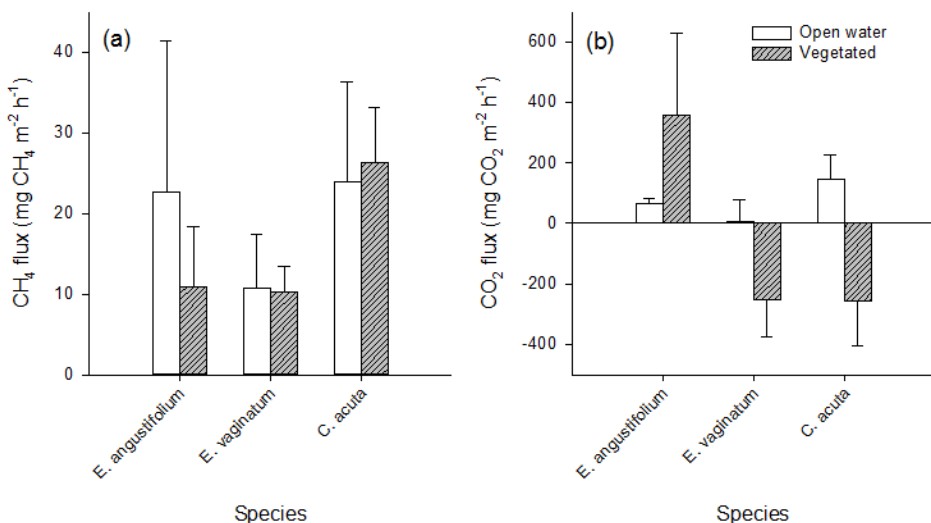

Figure 1. In situ paired-plot gas fluxes showing mean a) methane and; b) carbon dioxide fluxes at the field site in Abisko, Northern Sweden with standard error.

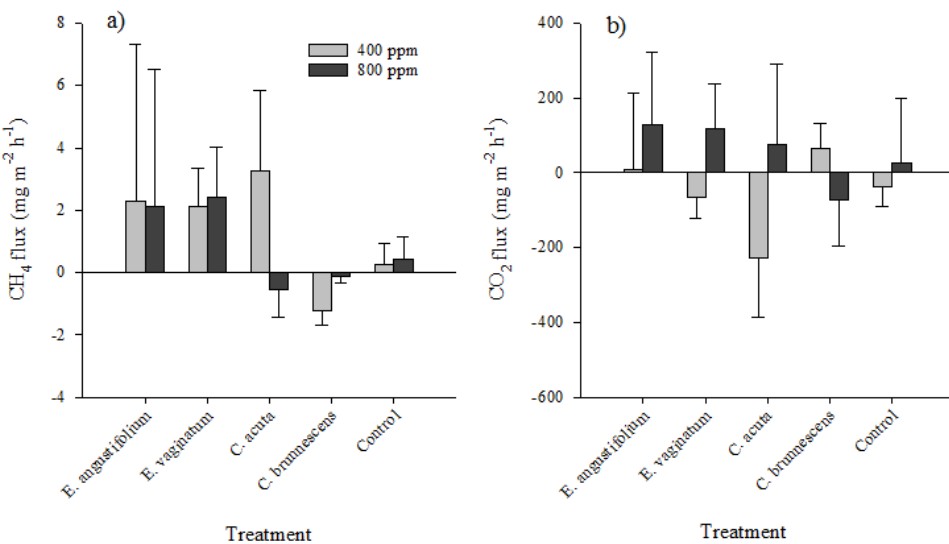

Figure 2. Mean experimental gas fluxes across five planted and unplanted treatments and under 400 and 800 ppm atmospheric $CO_2$ treatments showing a) methane and; b) carbon dioxide per unit area with standard error.





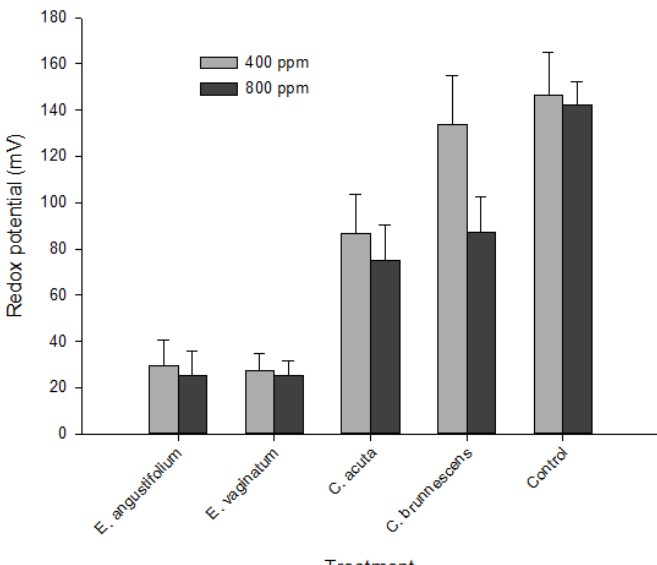

Figure 3. Mean redox potential with standard error for planted and unplanted treatments across 400 and
800 ppm atmospheric $CO_2$ treatments.





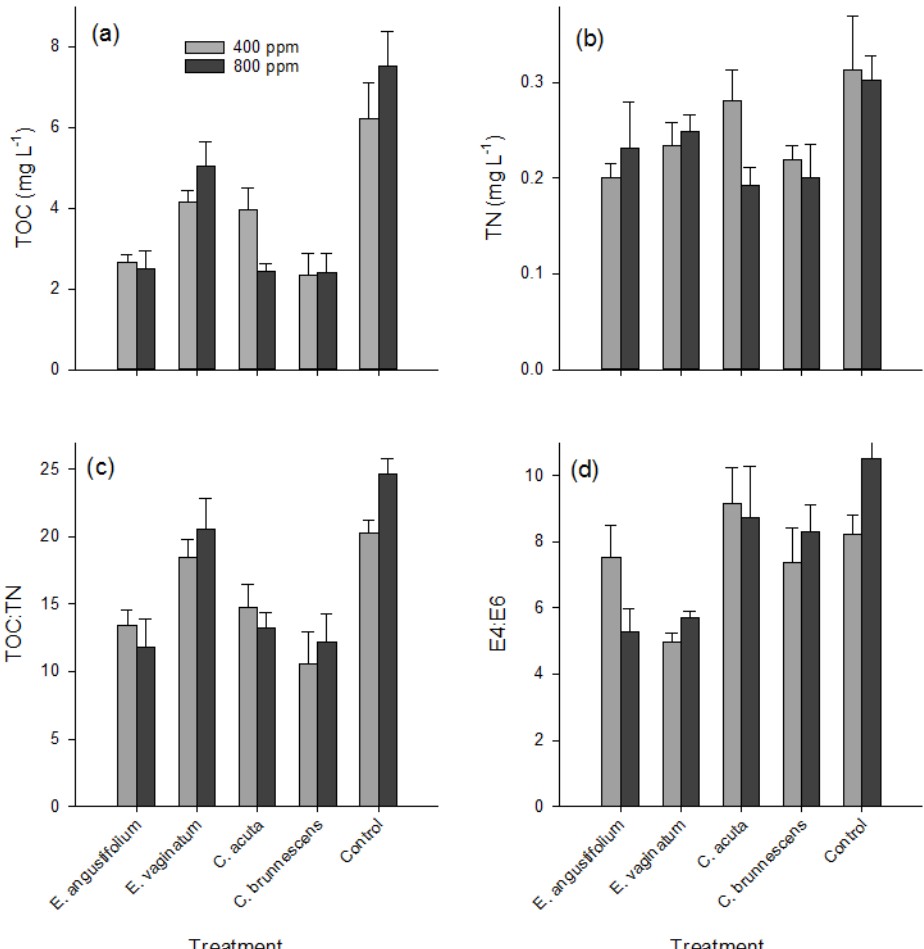

Figure 4. Means with standard error for a) Total Organic Carbon; b) Total Nitrogen; c) TOC:TN ratio and; d) E4:E6 ratio in pore water samples from time point 5 in experimental growth period for all Cyperaceae species plus control pots under atmospheric $CO_2$ conditions of 400 and 800 ppm.


Table 1. Mean total, above and below ground biomass and shoot:root ratio for all Cyperaceae
species across atmospheric $CO_2$ treatments with standard error.

| Species | Atmospheric $CO_2$ (ppm) | Total biomass (g) | Above ground biomass (g) | Below ground biomass (g) | Shoot:root ratio |
|---|---|---|---|---|---|
| *E. angustifolium* | 400 | 8.0 ± 1.9 | 2.4 ± 0.7 | 5.6 ± 1.2 | 0.39 ± 0.06 |
|  | 800 | 7.5 ± 1.7 | 3.8 ± 0.9 | 3.7 ± 0.9 | 1.10 ± 0.14 |
| *E. vaginatum* | 400 | 7.5 ± 0.7 | 2.6 ± 0.3 | 4.9 ± 0.6 | 0.57 ± 0.05 |
|  | 800 | 9.6 ± 1.4 | 4.0 ± 0.6 | 5.6 ± 0.8 | 0.73 ± 0.06 |
| *C. acuta* | 400 | 11.6 ± 1.4 | 3.6 ± 0.5 | 8.0 ± 1.1 | 0.51 ± 0.09 |
|  | 800 | 8.5 ± 1.1 | 3.4 ± 0.4 | 5.1 ± 0.7 | 0.67 ± 0.05 |





| | | | | | |
|---|---|---|---|---|---|
| *C. brunnescens* | 400 | 2.2 ± 0.4 | 0.7 ± 0.1 | 1.5 ± 0.3 | 0.51 ± 0.04 |
| | 800 | 3.7 ± 0.6 | 1.2 ± 0.2 | 2.5 ± 0.4 | 0.48 ± 0.04 |