# Peer review of "Ecosystem feedbacks from subarctic wetlands: vegetative and atmospheric CO2 controls on greenhouse gas emissions"

_Biogeosciences, 2016_

## Referee Comment (RC1) · Anonymous Referee #1 · 11 May 2016

The authors of the manuscript attempted to model how increasing carbon dioxide concentration in the atmosphere influence on methane emission using species-specific approach. The paper definitely falls within the scope of the journal. Coupling field measurements and laboratory experiments is a strong advantage of the paper extending our understanding of processes at different scales. While I find the MS very interesting, I feel that its quality could be improved. In particular, more consistency and clarity is needed. In addition, the gas measurement description and the discussion should be improved and extended.

General comments 1. Introduction is well structured and consistent. However, we recommend explaining importance of your exact study, not only of the general subject.

[Figure]

For example, is this the first study describing experiments with species from these latitudes and wetland types? Have any experiments been made earlier with species if studied families and genera? Why experiments with these species and from these wetlands are of special importance? In addition, we suggest explaining background of $CO_2$ and methane fluxes field measurements. Did you have any preliminary hypothesis how to connect field and laboratory experiments? Please, also mention it in Discussion.

2. We recommend adding schemes or photos concerning your experiments to make them clearer for readers. For example, you may add photo of chamber during measurement process. Now it is not clear if they were floating or standing on any collar. You should mention how you prevented disturbance of wetland soil during measurements (e.g. using bridges or a boat). Please, add scheme or photo of grown room experiment. Now it is not clear what the volume of samples was, how they were set into pots, what depth the water level has inside of the pots, what exposition conditions were, etc. Without detailed description, your experiment would be obvious only for small group of scientists closely related to this subject. Scientific community would benefit if your study were described more clearly.

3. Since methane cycle in wetlands is complicated, it includes different biological and physic-chemical interactions. Could you add a paragraph about your experiment limitations? For example, bubble transport in situ differs from the one in the laboratory experiments because in the last case the wind is absent. Changing in bubble transport influences other gas transport pathways. It is important because bubble transport is high in ecosystems with water level higher than the soil/peat surface. Volume of pores filled by air is of special importance too, as it was shown by many mathematical models. Probably, this volume had been changed during the experiment with further impact on other processes. If the air volume in pores was small, methane was not oxidized and rapidly emitted into the atmosphere. Is it possible to estimate bulk density changes of peat? In addition, such peat water properties as pH and ion concentrations could be changed during the experiment (see specific comments).

Specific comments

Site description

l. 105. Probably it would be more informative if mean July temperature would be reported instead of mean summer temperature.

l. 107. Can you explain what does "mesic" mean? It would be very useful for readers if more information about study site will be reported in Site Description. What are the typical values of pH and electrical conductivity for this mire? What is the trophic state of this mire, source of water supply (ombrotrophic or minerotrophic)? Does this object representative as subarctic wetland for both regional and global scale?

Experimental design and analysis. Field campaign

l. 126. It is not clear, what was the time of the whole time of flux measurement in the field? It is also important to say in this section, what was the typical level of uncertainty for individual flux measurements (both in the Lab and in the field). Please, provide this information.

l. 129-130. What was the precision of gas chromatography? It would be better to provide more details about gas chromatography (see for example (Repo et al., 2007)).

Repo, M. E., Huttunen, J. T., Naumov, A. V., Chichulin, A. V., Lapshina, E. D., Bleuten, W., & Martikainen, P. J. (2007). Release of $CO_2$ and $CH_4$ from small wetland lakes in western Siberia. Tellus B, 59(5), 788-796.

l. 132. What were the results of this examination? Do you use any criteria to check linearity?

Growth room experiments

l. 141. Can you say, what was the typical volume (or sizes) of peat samples, taken from mire. And please, clarify, whether these samples were taken with or without plant.

[Figure]

l. 143. Please, clarify, when WTL > 0, whether it means that water table is above or below mean peat/moss surface? Root-to-shoot ratio is very complexed parameter. It reflects a lot of influences and especially the influence of plant competition between each other. Can you say and proof somehow that plant density (on m2) was the same in your experimental pots and in the field? How many (typically) individual plants were grown in one pot?

l. 145. In the MS it is not reported that peat samples in the pot were submerged. I guess they were. Please provide information what was the height of water column (above peat sample) in pot and what was the typical volume of peat in the pot. I think, it would be very helpful for reader if you give photo of pot with a submerged peat sample, and also photo with a native sample, taken from the mire. Please, add it. It can be suggested that peat properties (e. g., pH and ion concentration) can strongly drift from natural values during long laboratory experiments. Of course it can influence on methane production and oxidation. Do you control them? Can you discuss it somehow in paper?

Discussion

In general discussion seems a little fragmentary and inconsistent. Probably it would be helpful to present a scheme showing all kinds of interactions and their results.

l. 246. Please provide an explanation, why methane fluxes on your site are so high. Does your mire also minerotrophic? If it is not probably it is better to compare with similar mires.

l. 252. Can you say what is "open water"? Does it have moss or peat bottom? Do open area and vegetated area have the same bottom properties. Does vegetation create a small submerged tussock? All these factors are important for methane emission.

l. 252-258. Even if sites are close to each other production is not the only thing that influence on plants. Plants are responsible not for 100% of organic matter supply for

methanogens. You are right that surface ch4 oxidation can unlikely be a reason of different emission. But deep ch4 oxidation (using oxygen transported by plants) can, and it should lead to difference in emission between vegetated and open areas. Then, as you mentioned, CH4 can be transported to plant tissues. And it also should lead to difference in emission between vegetated and open areas. Probably in this situation all these influences seems to be confounded. Thus we suggest 1) to list all possible plant effects on net methane emission; 2) to show that they have both positive and negative influence on net methane emission; 3) State that all these influences can compensate each other.

l. 262. Please, change "temperature" on "temperate".

l. 259-274. In this paragraph you list a number of studies reported different responses. It is useful. But all these responses are given for plant species which are different from species in your experiments. Can you compare your results for the same species? For the species from the same family or with the same characteristics? In the last sentences you state that there are fundamental properties, governing response of species to increased co2 concentrations. Does your study in agreement with these findings?

---

## Referee Comment (RC2) · Anonymous Referee #2 · 26 May 2016

The manuscript presents a combination of field measurements and a laboratory experiment aimed at discovering species-specific influences on subarctic wetland methane emissions under ambient and elevated atmospheric CO2 concentration. The study concludes that species-specific changes in biomass under elevated CO2 result in corresponding increases or decreases in CH4 emission.

The topic is of strong interest, as arctic and subarctic wetland methane exchange is known to be species-specific, and large uncertainties exist as to the response of northern latitude CH4 emissions to a changing climate. The study is well within the scope of Biogeosciences, and I think it has a lot of potential. The combination of in situ measurements and a laboratory experiment is a nice approach. The statistical approach

is valid. It is well-structured and the results are presented in a logical fashion. How-
ever, I have some concerns about the presentation and interpretation of the results,
and some recommendations to improve the clarity of experimental setup and breadth
of the discussion prior to publishing the paper.

Major points:

1. The abstract (lines 20-26), results (lines 206-210), and discussion (lines 331-333)
assert that changes in $CH_4$ emission followed changes in biomass under $CO_2$ fertiliza-
tion treatment, going on to cite species-specific directional changes. However, at least
half of the cited directional changes appear non-significant in Fig. 2a (i.e. E. angusti-
folium and E. vaginatum) as a result of a small mean effect relative to very large error
bars. The same generally be said of the biomass results for these species (Table 1).

Perhaps the underlying concern here is that the within-species variability is so large
that it stymies the reader's ability to interpret the differences both among species as
well as species-specific treatment effects. This could simply be an issue with the use
of bar graphs that lump all within-species variability (over time and replicates), making
it appear that there were very few significant differences.

I recommend playing with different plotting formats that more clearly show the signif-
icant effects identified in the mixed model. In addition, pairwise comparison statistics
could be used to support which species were actually responsible.

If the issue is not simply a matter of plotting, I recommend expanding the analysis to
look deeper into the within-species variability, both for the in situ and ex situ obser-
vations. Perhaps clearer species-specific influences will emerge after accounting for
other controls.

2. The intro states that vegetation is a primary control of $CH_4$ emission from wetlands,
and lists a few example mechanisms (line 55-64). While these examples are certainly
relevant to the present study, they are inadequate given that the study is centered on

species-specific plant influences on CH4 emission. For example, there is no mention of the widely-cited positive effect that vascular plants have on CH4 emission by providing a conduit for CH4 to the atmosphere (Morrissey and Livingston 1992, JGR-A; Joabsson et al. 1999, Trends in Ecology and Evolution; Joabsson and Christensen 2001, GCB; Christensen et al. 2003, Biogeochemistry; von Fischer et al. 2010, JGR; Kutzbach et al. 2004, Biogeochemistry). Many of the references above have focused on Eriophorum and Carex species. More background on species-specific CH4 control mechanisms (if possible, specific to the species investigated) will help set up the interpretation of the results in the discussion.

3. The Methods section is lacking in important details of the site layout and measurement setup, which adds difficulty in interpreting the results. Specifically:

a. Line 123: How far apart were the plots? How were they selected?

b. Line 123-126: How were the chambers placed upon the surface? Were collars used? Did this differ between vegetated and open water plots? How was access to these plots established to minimize disturbance?

c. Line 127-128: What time of day were measurements taken? What was the total time taken to measure all plots?

d. Line 131-132: Were any measurements rejected due to non-linearity?

e. Line 137-138: What were the dimensions of these chambers?

f. Line 141-142: Are these replicates per 400 or 800 ppm treatment? Or total?

g. Line 142-144: What was the water level in the pots? Was water from the site used? What were the dimensions of the transplanted peat/plant, and were the same dimensions used across the experiment? Why were the recovered plant and soil samples separated? I would expect them to remain whole to minimize shock.

h. Line 146: If light levels were constant during daytime, this sentence should read

"...daytime light levels were constant at 400..."

i. Line 154-155: Were measurements taken under "daytime" conditions?

j. Lines 156-158: Why were only 2 air samples made to estimate flux? This would prohibit checking for linearity.

k. Lines 161-163: Please specify manufacturer and model of redox probe and rhizon samplers.

l. Line 168-170: Was initial starting biomass estimated? It does not appear so, since biomass estimates were measured destructively only at the end of the experiment. This is fine, but the manuscript interprets the biomass results (Table 1) as if the starting biomasses were equal for all same-species replicates across 400 ppm and 800 ppm treatments. This assumption and justification for making it should be explicitly stated, since the main conclusion of the paper stems from these measurements.

These missing site and experimental details are most apparent when attempting to understand the order of magnitude difference in CH4 emissions between the in situ and ex situ observations (Fig. 1 vs. Fig. 2). Reconciling these differences requires details on the dimensions of transplanted plants. I would expect the in situ and ex situ flux measurements to mimic each other as much as possible in terms of filling the floor area of each pot so that a per-area flux measurement would be comparable. If not, the reader needs to be oriented to expected differences with explanation and justification.

4. There is no mention of the plant extension growth measurements in the results. These data would be useful to support the destructive biomass sampling results performed at the end of the experiment.

5. The ex situ and in situ results are presented and discussed in isolation. Is there anything to be learned about the in situ results from the ex situ results, and vice versa?

6. The TOC, TN, TOC:TN, and E4:E6 are hard-won ancillary data that could potentially explain within-species variability as well as species-specific responses to elevated

CO2. However, aside from a general discussion (line 304-311) on how this data "highlights the influence of species composition in these wetlands on rhizospheric carbon inputs, ..., with implications for CH4 production", the implications for different CH4 production between species and associated response to elevated CO2 is left untouched.

7. The contrasting changes in ex situ fluxes over time (line 206-210) is an important result that may shed light on species-specific plant-mediated CH4 transport, yet it is unmentioned in the discussion.

Minor points:

8. I am unconvinced by the explanation (lines 251-257) as to why open water and vegetated CH4 emissions were not significantly different, given that the dominant transport pathway of CH4 to the atmosphere often differs substantially between vegetated (transport through vascular plant material) and open water areas (diffusion/ebullition). More discussion that references transport pathway is warranted here, since diffusion alone is unlikely to support the very high CH4 fluxes observed at this site.

9. What is the future direction of this research? What are the limitations of this study and unanswered questions to be resolved?

10. The manuscript would benefit from editing to correct several typos and grammatical errors.

---

## Author Comment (AC1) · 4 Jul 2016

We have responded to the reviewers major and minor points in order.

1. We agree with the reviewer that it is a good idea to develop the rationale of the study further. Indeed it is the first study of impacts of elevated CO2 in subarctic peatlands. We chose the species we chose because both Eriphorum and Carex are known to contribute to CH4 production in these wetlands (the relevant references from the Introduction is Prater et al., 2007; Christensen et al., 2004; Hodgkins et al., 2014). For the revisions we plan to provide more depth in this section in line with the reviewers' suggestion referring to the role of these species in the CH4 cycle. Regarding our working hypothesis for the field measurements, our hypothesis was that we would see

more CH4 emitted from plots with plants due to root exudation stimulating CH4 production and plant mediated transport. We also hypothesised more CH4 being emitted from higher biomass species. We are happy to add this to the paper and develop this aspect of the work in the discussion as suggested by the reviewer. It will also help integrate the laboratory and field data as suggested. 2. In the field we used floating chambers. As the wetlands were shallow we did not need to use boats or bridges. To reduce the impact of disturbance plots were allowed to settle for 5 minutes after the chamber was put in place before the first gas samples were taken. We assessed all of the regressions for linearity and in a few cases we had to remove a set of samples when high initial CH4 concentrations indicated ebullition during chamber installation. We'll describe experimental method including a schematic diagram of the field and laboratory set up as supplementary information together with photos to clarify the sampling conditions. In the pot experiment the peat was covered by 2-3 cm of water. 3. In response to this valid point by the reviewer we suggest addressing the limitations of the experimental set up in the discussion section where we can directly link the limitations of the respective field and laboratory conditions to the interpretation of the data. It is worth noting that the field experiment was intended to be complementary to the laboratory experiment not be used as a direct comparison due some of the factors the reviewer points out. In the discussion we can describe the limitations of comparing them directly in terms of difficulties associated with long term mesocosm experiments and the impacts on soil and water chemistry, differences associated with gaseous exchange mechanisms including lack of circulation resulting from weather factors etc. It is possible that the structure of the peat has changed during the experiment as the reviewer suggests however we don't think this will have had a major impact on methane oxidation as the peat in the laboratory experiment was always submerged. It is however possible that methane oxidation rates were affected by root growth and root oxygen release during the cause of the experiment. Specific comments Line 105. We will add the mean July temperature from the nearby Abisko weather station. Line 107. We will add information on the pH and electrical conductivity and can add in the actual moisture levels of mesic

areas (i.e. peat of intermediate moisture). The pH, electrical conductivity, active layer depth and vegetation in our study mire is representative of large areas of palsa mires in northern Scandinavia. Line 126. In the field fluxes were measured after 5, 10, 20 and 30 min with a total incubation time of 30 minutes. The R2 values of the regressions was variable depending on what process dominated therefore we chose not to use a hard cut using R2 (e.g. in the instance of low CH4 production and CO2 fluxes close to zero indicating a balance between uptake and emission) instead we visually inspected the regressions to ensure they were not affected by extreme values. Line 129-130. We will report the analytical error and the specific GC settings in the manuscript in line with the recommended reference. Line 141. Peat was taken as bulk samples (several ca 20*20*20 cm blocks), and separated into pots. Peat was taken one metre below water surface in areas which were free from vegetation. Line 143. The water level was 2-3 cm above the peat surface. Due the growth form of E. vaginatum these were measured and grown as tussocks in the field and in the lab, the other species were measured as individual shoots both in the field and the laboratory. We will clarify this further in the text in the methods section as this has important implications for comparing the field and laboratory data as the reviewer points out. Line 145. Approx. 1 litre of peat, which was submerged as explained above. We did not control pH and ion concentrations but measured water chemistry at the end of the experiment and redox throughout. We suggest discussing these limitations in the discussion as mentioned above. Line 246. The site is minerotrophic and comparable to the sites referred to in the discussion both with regards to comparable basic chemical parameters and vegetation composition: The site is affected by water input from snowmelt run-off from surrounding areas, it is also influenced by influences by ground water. The pH is 4.3 $\pm$ 0.1 and the electrical conductivity is 66 $\pm$ 31 uS, extractable PO43- is 3.9 $\pm$ 1 ug g-1 and NH4+ is 0.12 $\pm$ 0.02 ug g-1. We suggest to add these data to the paper to inform the readers about the peat properties. The high emissions are comparable to nearby mires which have similar properties e.g. melting permafrost causing subsidence, flooding and establishment of graminiods which support high methane emissions (e.g. Prater et al., 2007,

Hodgkins et al., 2014). We will make this more explicit in the text of the revised ms. Line 252. All the sampling sites were underlain by peat, at relatively similar stage of decomposition (fairly shallow mire). E.vaginatum is tussock forming but not the other species. Line 252-258. We agree that root oxygenation is an important process which may impact on emissions. We will be happy to take your suggestion on board and in the resubmission we propose to outline plant impact on $CH_4$ emissions, the direction (positive or negative) or the impact and potentially antagonistic influences. Line 262 We will make this change. Line 259-274. We have not identified any studies on elevated $CO_2$ on the common boreal/subarctic wetland species we chose in our study which adds novelty to the study but makes comparisons with other studies more difficult. However, these Eriophrum and Carex are relatively well studied with regards to their role for plant mediated transport of $CH_4$ (as pointed out by reviewer 2) which we plan to outline in the introduction in response to the reviewers comments. Our last sentence of that paragraph referees to a study which found that nutrient limitation impacts on the plant growth responses to elevated $CO_2$. However, in our study we did not analyse the nitrogen demand of these species so it is difficult to assess if the response we found are driven by nitrogen demand specifically.

---

## Author Comment (AC2) · 4 Jul 2016

We have added the response to the reviewers comments in order below:

1. In response to this point we have plotted the biomass data (currently in table 1) as barcharts which mirrors the format for the figures showing the CH4 and CO2 fluxes and well as the CH4 treatment responses making the link between biomass and CH4 flux more obvious. We will add SEDs to the figures to aid the readers to compare treatments as requested by the reviewer. We have also run additional statistics as suggested by the reviewer and we propose to include a set of scatter figures illustrating the positive relationship between the biomass parameters and the CH4 fluxes. We can show a significant positive regression between total biomass as well as root biomass

and CH4 fluxes (R2>0.5). 2. This comment chimes with reviewer 1s comment about adding more detail about the different ways in which plants affect CH4 emissions (discussion l252-258). We will add further detail to include wider factors and mechanisms like conduit transport, although we dismiss this as a major influence on gas transport due to our field results. We can include more background on species specific factors affecting these relevant mechanisms 3. Add further detail to experimental method, we have given the additional detail requested by the reviewer below and will include this information in the revised ms. a) paired plots were 1-2 meters apart with sampling blocks spaced ca 100 m apart. b) chambers were floating both for vegetated and open water plots. All plots had standing water. c) measurements were taken between 10 am and 5 pm. Note that sampling coincided with the period of midnight sun at this latitude. Plots were blocked (one block consisted of six plots, plots with the three target species and adjacted paired open water areas, total of 30 plots) to account for changes in weather conditions and plant activities over time. All plots were sampled twice over a 5 day periods, with the first set taking two and a half day and the second sampling occasion also taking two and a half day. d) in a few instances measurements were rejected due to non-linearity e) the dimensions of chambers are detailed in the experimental design section f) the number of replicates are per treatment g) water level in pots were less than 2-3 cm above peat surface. We used tapwater water to adjust the water levels through-out the experiment. We used the same volume of the peat in the pots. Peat was taken as bulk samples, and separated into pots around the plant roots to good contact between the plant roots and surrounding peat. We acknowledge this process disturbed the peat structure. Indeed also the plants received a transplant shock as leaves and roots needed to be trimmed back before transport from the wetland site. However, the plant-soil pots for all species and CO2 treatments were prepared in the same way to allow comparisons among treatment. Peat was taken one metre below water surface without plants. Due to the disturbance and other experimental artefacts (e.g. relating to plant densities as pointed out by reviewer 1) care needs to be taken when interpreting the data and our laboratory fluxes cannot be directly translated to the

field conditions. We suggest outlining these limitations more explicitly in the discussion whilst improving the connection between the field and laboratory data. h) Thank you we will amend this. i) all measurements were taken under 'daytime' conditions j) this was a compromise to allow for all of the gas samples from the 62 pots in the two growth rooms to be collected over two days. We did not see any signs of ebullition events during sampling (i.e. no gas samples with very high CH4 values) likely because most of built up bubbles were release from the low density peat both due to the slight disturbance of the peat during watering and because the sample pots were moved around in the growth rooms prior to fitting the head spaces. It is possible that there was some bubble release during sampling in the laboratory experiment resulting in an over estimation of the fluxes. Note that we allowed the mesocosms settle for ca 10 min before the head space was fitted on the plant-soil pots and after the head space was gently fitted on the pots they were left for another 10 min before the first samples were drawn. k) we will add this information to the revised manuscript. l) we will clarify that biomass was not equal at the start of the study. To account for the different sized plants at the start of the experiment we paired them according to size (within each species) and allocated the treatment (400 or 800 ppm) randomly between the plants in each pair to ensure difference in initial starting biomass was accounted for in the experimental design. We will specify more clearly the differences in plant densities in the field and laboratory measurements in the methods section. To aid comaparison between field and lab measurements the field flux measurements these were made over individual shoots for E. angustifolium and C. acuta (as their growth form allowed this) while the for the tussock forming E. vaginatum the field measurements were made over small hummock, aiming to replicate the size of the clumps of E. vaginatum used in the lab. 4. We will add in the extension growth biomass data to the paper to support the treatment effects: The impact of the treatments on extension growth is significant with consistently lower extensions growth in the 800 ppm treatment (it is worth noting that the extension growth data is more consistent than the above ground biomass with respect to treatment effects probably due to the issues linked to "starting " biomass,

see reviewer 2's point 3l and our response above). This reflects the reduced $CO_2$ sink strength in this treatment. We will develop this link between growth responses and the $CO_2$ uptake in the result and discussion. 5. With regards to linking laboratory and field data there as some links which we propose to develop in response to the reviewers' comment: First the species which act as $CO_2$ sinks in the field also acts as $CO_2$ sinks in 400 ppm treatment the laboratory experiment while the species that is a $CO_2$ source in the field was also a weak $CO_2$ source in the 400 ppm treatment in the laboratory suggesting that the laboratory control condition to some extent reflect the field conditions. Second the fact that the plant treatments actually lower peat redox conditions is interesting the context of the reviewer 1s' point about deep $CH_4$ oxidation (l252-258) we agree with the reviewer that plant root oxygen release with increase $CH_4$ oxidation in the rhizosphere. However, the fact that the planted treatments had lower redox that the unplanted peat at the end of the laboratory experiment suggest that stimulations of reducing processes due to the presence of plant roots (e.g. release of labile substrates for decomposition) may actually increase the potential of $CH_4$ production in the rhizosphere. We propose to bring these points into the discussion. 6. We agree with the reviewer that we can make better use of this data. The points we suggest to bring into the discussion are: The corresponding patterns with lower root biomass, TOC and TN concentrations and lower $CH_4$ emissions in the 800 pmm C. acuta treatment suggests a link between root biomass, root exudation and $CH_4$ fluxes. As different plant species (we will specifically refer to gramiods and our study species using information from existing studies) allocate C differently and also differ in the amount and composition of their root exudates. We propose to develop this line of in query in the discussion as a potential explaintion to the species-specific responses with regards to the $CH_4$ fluxes to the elevated $CO_2$ treatment. Additionally the lower redox in the 800 ppm C. brunnescens treatment may explain/contribute to the diminished $CH_4$ sink in the 800 ppm treatment for this species. We speculate that the lower redox in the 800 ppm treatment of this species may be due stimulation of reducing processes in the rhizosphere due to the greater root biomass in this treatment. In future research it would be interesting

to explore the role of biomass for producing reducing condition as this contrasts to our current understanding of roots impact on soil redox conditions under water logging. 7. We agree with the reviewer that the differences in the temporal pattern in the laboratory fluxes are interesting. Our interpretation of the data is at the greater initial $CH_4$ emissions from the C. acuta is linked to the more rapid growth of this species at the start of the laboratory experiment (highest extension growth rates and also high $CO_2$ release at the start suggesting high activity in the rhizosphere). In parallel we speculate that the greater $CH_4$ release at the end of experiment for E. angustifolium reflects the build-up of biomass over time. It is not quite clear us how the reviewer thinks this information should be used to discuss species specific effects on plant-mediated $CH_4$ transport as in our laboratory experiment we are not able to separate emissions through the plants from other emissions pathways and in the field measurements we did not detect any differences between open water and vegetated areas with regards to $CH_4$ emissions. We propose to outline the parallel between growth rates and $CH_4$ emissions in the discussion but refrain from speculating as to how the laboratory study may inform our understanding of species specific plant mediated $CH_4$ transport in the field as we feel that the data we have is not strong enough to under pin a robust discussion. 8. We will make this section more nuanced and refer to studies exploring different emission pathways and the different way plants may impact emissions of $CH_4$ to account for this comment by reviewer 2 and also reviewers 1s' point about deep methane oxidation. 9. There are two main questions raised by this research both of which the reiwevers has touch upon: First what does our findings mean in the field context and over long time periods? Second is what plant traits is driving the different below ground biomass responses (below ground biomass was a strong predictor of $CH_4$ emissions) among the plant species? These two questions needs answering before plant mediated impacts on $CH_4$ emissions in a $CO_2$ rich world can be predicted. 10. We will ensure the paper is proof read before submission on the revised manuscript.